

**Ionospheric influence on the seismo-telluric current related to electromagnetic**
**signals observed before the Wenchuan $M_S$=8.0 earthquake**

4              Mei Li[1,2], Handong Tan[2] and Meng Cao[2]

⑴ China Earthquake Networks Center, China Earthquake Administration, No.5,

7       Sanlihe Nanhengjie, Xicheng District, 100045 Beijing, China.

⑵ China University of Geosciences, No.29, Xueyuan Road, Haidian District, 100083

9       Beijing, China.

Corresponding author: Handong Tan, China University of Geosciences, No.29,
Xueyuan Road, Haidian District, 100083 Beijing, China. (thd@cugb.edu.cn)
**Abstract.** A three-layer (Earth-air-ionosphere) physical model, as well as a
two-layer (Earth-air) model, is employed in this paper to investigate the ionospheric
effect on the wave fields for a finite length dipole current source co-located with the
main fault of an earthquake when the transmitter-receiver distance is up to one
thousand kilometers or even more. The results show that all electrical fields are free
of the ionospheric effect for different frequencies in a relative short range, e.g., ～300
km for f=1 Hz, implying the ionospheric influence on electromagnetic fields can be
neglected within this range that becomes smaller as the frequency increases. However,
the ionosphere can give a constructive interference to the waves passed through and
make them decay slowly when an observation is out of this range and the ionosperic
effect can be up to 1-2 magnitudes of the electrical fields. For an observed 1.3 mV/m
signal at 1,440 km away for the Wenchuan $M_S$=8.0 earthquake, the expected
seismo-telluric current magnitude for the Earth-air-ionosphere model is of $5.0 \times 10^4$
$kA$, which is of one magnitude smaller than the current value of $3.7 \times 10^5$ kA obtained
by the Earth-air model free of ionospheric effect. This indicates that the ionosphere
facilitates the electromagnetic wave propagation, as if the detectability of the system
is improved effectively and it is easier to record a signal even for stations located at
distances beyond their detectability threshold.
**Keywords.** Ionospheric influence on electromagnetic waves; The Wenchuan
earthquake; Seismo-telluric current



## 1 Introduction

The fact that Electro-Magnetic (EM) emissions accompany every stage of large earthquake preparations seems undebatable although short-term earthquake prediction is still one of the most challenging targets in Earth science today (Eftaxias et al., (2002). Meanwhile, the Ultra-Low Frequency (ULF) band is of particular interest because only EM signals in the ULF range and at lower frequencies originated in the Earth's crust can be easily recorded at the Earth's surface without significant attenuation comparing with 'high' frequency emissions that might be emitted at epicenter depths at more than 10 km, even several hundreds of kilometers. Recently, ULF electromagnetic anomalous phenomena related to strong earthquakes have been investigated and reported in increasing numbers. Two notable examples are the Loma Prieta $M$s=7.1 earthquake on October 17, 1989 (Fraser-Smith et al., 1990; Bernardi et al., 1991), as well as the Spitak $M_S$=6.9 earthquake on December 7, 1988 [Molchanov et al., 1992; Kopytenko et al., 1993). It was found that, these two earthquakes have very similar electromagnetic emission fluctuation characteristics. The intensity of the signals began to increase 3-5 days before the earthquake at Spitak and 12 days before that at Loma Prieta. The most important point is the electromagnetic emissions exhibited a maximum 4 h before the Spitak event (f=0.005-1 Hz, D=200 km, A=0.2 nT) and 3 h before the Loma Prieta event (f=0.01-10 Hz, D=7 km, A=1.5 nT).

A more recent example reported by Li et al. (2013) is the Wenchuan $M$s=8.0 earthquake on May 12, 2008, a typical mid-crust, which resulted in great devastation and 69,000 deaths. This earthquake was preceded by more than one month of increasing anomalous ULF emissions with a climax starting on May 9, three days before the Wenchuan main shock (f=0.1-10 Hz, D=1,440 km, A=1.3 mV m-1).

Many simulating rock-pressure experiments were carried out in order to understand the producing mechanism of the electromagnetic information associated with seismic activities. Laboratory experiments by Qian et al., (1996; 2003) and Hao et al. (2003) demonstrate that, electromagnetic signals are always recorded when rock samples are subjected to dynamic stresses. There is a close relationship between the produced signal and the formation of micro-cracks in the rock. Furthermore, the climax of the signal occurred when the main rupture happened and the magnetic pulses of shorter-period appearing at the last stage of the experiment may be induced by instantaneous electric current of the accumulated charge during the cracking acceleration.Strong electromagnetic signals are generated while rock samples were



fracturing (Panfilov, 2014). As rocks upon stressing, dislocations are generated that
sweep through the mineral grains and break the peroxy links. The positive holes are
activated as charge carriers, and have the remarkable capability to flow out of the
stressed sub-volume and spread into the surrounding unstressed rocks. This situation
can be considered as analogous to a battery which generates electric currents along
the stress-gradient direction (Freund, 2002; Freund and Sornette, 2007; Freund, 2009,
2010). A gabbro sample (30×15×10 cm$^3$)from Shanxi, China, came into the test and
a 55 nA current recorded about 2 seconds before failure, with the load being at about
30,000 lbs and the maximum spike reaches 450 nA when the main failure took place
(Freund, 2009). Up to now, no clear explanation has been given although several
physical mechanisms have been proposed to interpret the generation of EM emissions
and electrical currents observed either during seismic activity or in the laboratory
experiments.   These   include   the   electrokinetic   and   magnetohydrodynamic,
piezomagnetism, stress-induced variations in crustal conductivity, microfracturing,
and so on (Draganov et al., 1991; Park, 1996; Fenoglio et al., 1995; Egbert, 2002;
Simpson and Taflove, 2005). Whatever the physical mechanism of electromagnetic
generation is, it is well established that, during rock experiments conducted under
laboratory conditions, a strong electrical current is produced when rocks are stressed,
especially at the stage of the main rupture. So, like what Bortnik et al. (2010) wanted
to know, what is the electrical current necessary to produce an observable magnetic
signal on the ground, at a given distance from the epicenter and for an assumed
ground conductivity? In their work, an infinitesimally short, horizontal dipole located
at a hypocenter depth in the half-space (Earth) is used to estimate the magnitude of
the seismo-telluric current required for the "Alum Rock" $M_W$=5.6 earthquake on
October 31, 2007. The observable electromagnetic ground signals (f=1 Hz, D=2 km,
A=30 nT) and the results show that for an observed 30 nT pulse at 1 Hz, the expected
seismo-telluric current magnitudes fall in the range ~10–100 kA.

Unlike a parameter of the "Alum Rock" $M_W$=5.6 earthquake (D=2 km), the

distance between the epicenter of the Wenchuan $M_S$=8.0 earthquake and the observing
station is D~1,440 km (Li et al., 2013), i.e. several times higher than the height of the
bottom of ionosphere (h~85-100 km) (Kuo et al., 2011; Cummer, 2000; Yamauchi et
al., 2007). When we investigate electromagnetic emissions induced by an electrical
current or a magnetic moment on the surface or beneath the Earth, the effect of the
medium air, crustal as well as ionosphere should be taken into account because of



these three media being of different conductivities and so we need to consider a
lithosphere-atmosphere-ionosphere electromagnetic coupling. The ionosphere plays
an important role in radio propagation at Extremely Low Frequency (ELF) and Very
Low Frequency (VLF), the ground and the ionosphere are good electrical conductors
and form a spherical Earth-ionosphere waveguide (Cummer, 2000). In addition, in the
magneto-telluric (MT) method, widely used in exploring petroleum or mine, the
ionospheric influence on electromagnetic (EM) fields should be considered when the
transmitter-receiver distance of a large-scale and large-power fixed resource is up to
one thousand kilometers. EM fields can be strengthened in the ionosphere as it is
shown when we use analytical solutions of Maxwell equations, as well as numerical
ones of the "Earth-ionosphere" mode with a source on the ground or in the air (Fu et
al., 2012; Li et al., 2010a; Li et al., 2010b; Xu et al., 2012; Li et al., 2011). Therefore,
comparing with an electromagnetic attenuation without ionospheric effect, the point is
to evaluate the ionospheric influence on the electromagnetic propagation when the
transmitter-receiver distance is up to one thousand kilometers or even more.
Furthermore, the comparison between the observation distance (D=1,440 km) and the
length of the Wenchuan earthquake main rupture L=∼150 km (Zhang et al., 2009)
indicates that the length of the dipole source is not negligible. So in this paper, using
the work of Key (2009), a three-layer (Earth-air-ionosphere) physical model, as well
as a two-layer (Earth-air) model, containing a finite length dipole current source
co-located along the fault and beneath the Earth is introduced in Sect. 2. For specified
parameters, some simulation results of the dipole source with and without ionospheric
effect are given in Sect. 3. In Sect.4, we define and limit our assumed parameter
values, present the results for the Wenchuan earthquake case. Discussion and
conclusions are given in Sect. 5 and Sect.6, respectively.

**2 Description of the modeling methodology**
In order to study the electromagnetic fields emitted by a long dipole current
source, the approach used here follows the magnetic vector potential formulation
described in Wait (1982) and developed by Key (2009), who generalized the
formulation to allow for multiple layers above the transmitter (in addition to multiple
layers below). He used exponential forms for the recursions rather than hyperbolic
functions in isotropic media, which consists of N layers of isotropic conductivity$\sigma_i$
where$i = 1, ... , N$, and which uses a right-handed coordinate system with the z axis





pointing down. Assuming a time-harmonic source with $e^{-i\omega t}$ time dependence,
negligible magnetic permeability μ variations, and angular frequencies ω that are
low enough so that displacement currents can be neglected, Maxwell's equations are
$$\nabla \times \mathbf{E} = i\omega\mathbf{B}, \qquad (1)$$
and
$$\nabla \times \mathbf{B} = \mu\sigma\mathbf{E} + \mu\mathbf{J}_s . \qquad (2)$$
Expression $\mathbf{J}_s = \mathbf{I}\delta(\mathbf{r} - \mathbf{r_0})$ is the imposed electric dipole source at position $\mathbf{r_0}$ with
vector moment $\mathbf{I}$, and here is restricted to an infinitesimal dipole with unit moment.

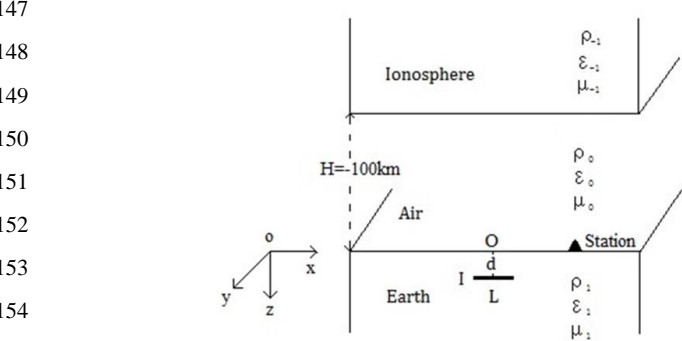

**Fig.1.** An x-directed dipole current source, with its central coordinate (0, 0, d), is placed in the
bottom medium (Earth) of a three layer modeling (Earth-air-ionosphere model), where z is defined
positive in the downward direction.
Based on the model set up by Key (2009), some modifications will be done in
this study in order to answer the questions illustrated above. A physical model is
specified. It has three layers, Earth, air and ionosphere, which is called
Earth-air-ionosphere model. Its coordinate system is denoted in Fig.1 with z-direction
being downward. An x-directed dipole of a length L and a current I is placed in the
bottom medium (Earth: $z > 0$), which is homogeneous and has the electrical
properties: magnetic permeability $\mu_1$, permittivity $\varepsilon_1$, and conductivity $\sigma_1$. The
middle medium (air: $-100\text{ km} < z < 0$) is described by its electrical properties $\mu_0$,
$\varepsilon_0(= 8.854 \times 10^{-12}\text{ Farad m}^{-1})$ and $\sigma_0(= 10^{-14}\text{ S m}^{-1})$ . The top medium
(ionosphere: $z < -100\text{ km}$) is characterized by electrical properties $\mu_{-1}$, $\varepsilon_{-1}$ and
$\sigma_{-1}(= 10^{-5}\text{ S m}^{-1})$.



As a comparison, a two-layer model (Earth-air model) including in Earth
medium (z > 0), as well as air medium (z < 0), is also established during the study.
All the corresponding parameters described are the same as these of
Earth-air-ionosphere model.
We assume that the total space is non-magnetic and that the magnetic
permeabilityμvariations are negligible in the different layers, i.e. $\mu_1 = \mu_0 = \mu_{-1} =$
$4\pi \times 10^{-7}$ Farad m$^{-1}$. However, the ionosphere as the electrically conducting
section of the upper atmosphere play a so important role on the electromagnetic
propagation that we set $\varepsilon_{-1} = 5\varepsilon_0$ when an ionospheric effect on electromagnetic
transmission is taken into consideration. On the same manner we have $\varepsilon_1 = \varepsilon_0 =$
$8.854 \times 10^{-12}$ Farad m$^{-1}$, i.e. $\varepsilon_1$ is not considered as zero during all calculations.
Under these conditions, the formula listed above are still suitable and more
explanations about the potential formulation of a horizontal electric dipole can be
found in the Appendix A of Key (2009) and related programs are available with an
access to the website (http://marineemlab.ucsd.edu/). The horizontal finite length
dipole source can be seen as an integral of an infinite small horizontal dipole during
related calculations.

**3 Simulation results**
According to these two models built above, several free parameters must be
specified in order to investigate the attenuation characteristics of the electromagnetic
fields emitted by a long x-directed dipole current source. As for the parameters of the
dipole current source, we select L=150 km, the Wenchuan earthquake main rupture
stage within 30 s out of 90 s (～300 km) based on Zhang et al., (2009, Fig.1), the
depth d =19 km (Xu, 2009), the hypocenter depth of the Wenchuan case and the
current is set to be I=1 A temporarily. Here, the Earth is considered to be an isotropic
media with an average conductivity $\sigma_1$, and we assume $\sigma_1 = 1.0 \times 10^{-3}$ S m$^{-1}$ at
this time, i.e. $\rho_1 = 10^3$ ohm · m, although the ground conductivity depends not only
on the local petrology, but also on the porosity, temperature, and pressure (e.g., Wait,
1966). All these parameters are common to two models. While the parameter
$\varepsilon_{-1} = 5\varepsilon_0$ is of most importance during the calculation in three-layer model in that it
potentially can affect the transmission of electromagnetic waves produced by the
dipole beneath the Earth, and possibly induces the Earth-atmosphere-ionosphere
electromagnetic coupling.




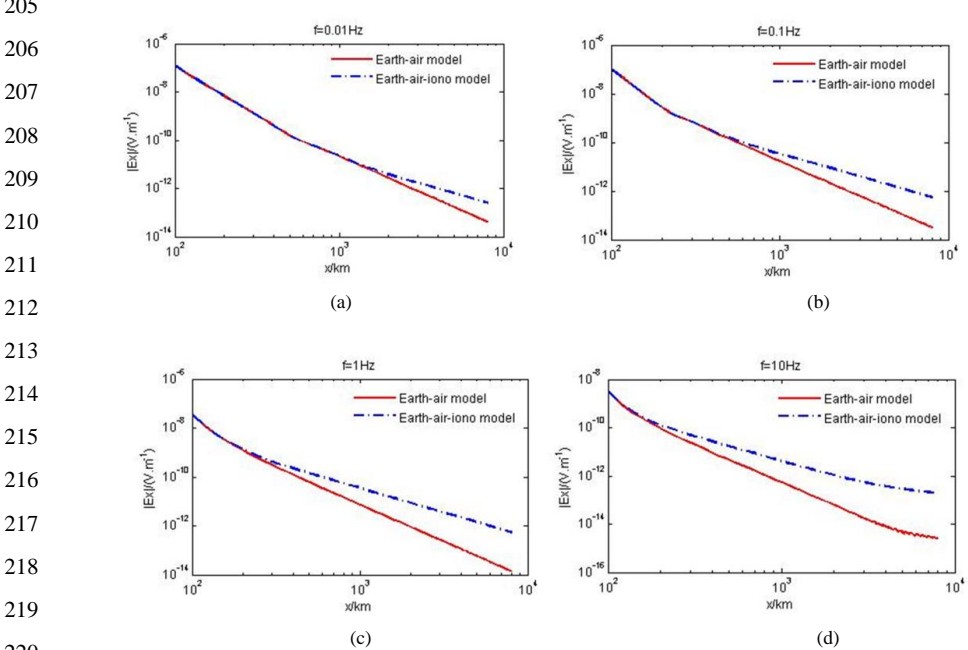

**Fig.2.** Electric field $|E_x|$ decay curves along x-axial direction as a function of the observing distance for the Cartesian coordinate system with different frequencies. Red solid lines stand for electric field curves for Earth-air model and blue dot lines denote electric field curves with the ionospheric effect for Earth-air-ionosphere model.

(a) Total $|E_x|$ for f=0.01 Hz;    (b) Total $|E_x|$ for f=0.1 Hz;

(c) Total $|E_x|$ for f=1 Hz;        (d) Total $|E_x|$ for f=10 Hz;

Fig.2a-d displays electric field component amplitude $|E_x|$ decay curves along the x-axial direction with the frequencies f=0.01 Hz, f=0.1 Hz, f=1 Hz, and f=10 Hz respectively for the Cartesian coordinate system up to ～10,000 km on the Earth's surface.

It can be seen from Fig.2a-d, first, the electrical field with "high" frequency has a big attenuation although all curves for both Earth-air model (red solid lines) and Earth-air-ionosphere model (blue dot lines) decay rapidly as the distance increases. Second, each group of curves run at the same level for one fixed frequency, e.g., f=1 Hz, when an observing point is located at a relative near distance, ～300 km for f=1 Hz (Fig.2c) for example. That is to say, the ionospheric influence on electromagnetic field transmissions can be neglected within this range. However this range changes



for different frequencies and it becomes smaller as the operating frequency of the
current source increases (e.g., more than 1000 km for f=0.01 Hz (Fig.2a) and only
~200 km for f=10 Hz (Fig.2d)). Third, the most important result is, as the distance
increases, field curves with an ionospheric effect (blue dot lines) run by a different
way from that of curves without an ionospheric effect (red solid lines) and the
ionospheric lines attenuate more slowly. Now, this kind of ionospheric influence
cannot be neglected at this time anymore. The ionospheric difference is about 1
magnitude ($\times$10) for all the frequencies listed and even once up to 2 magnitudes for
f=10 Hz within the range shown in Fig.2. For example, the ionospheric difference
value shows 1 magnitude from $\sim$840 km, up to 2 magnitudes from $\sim$3,700 km for
f=10 Hz (Fig.2d).

**4 The Wenchuan $M_S$=8.0 earthquake as a sample**
**4.1 Estimating the seismo-telluric current magnitude**

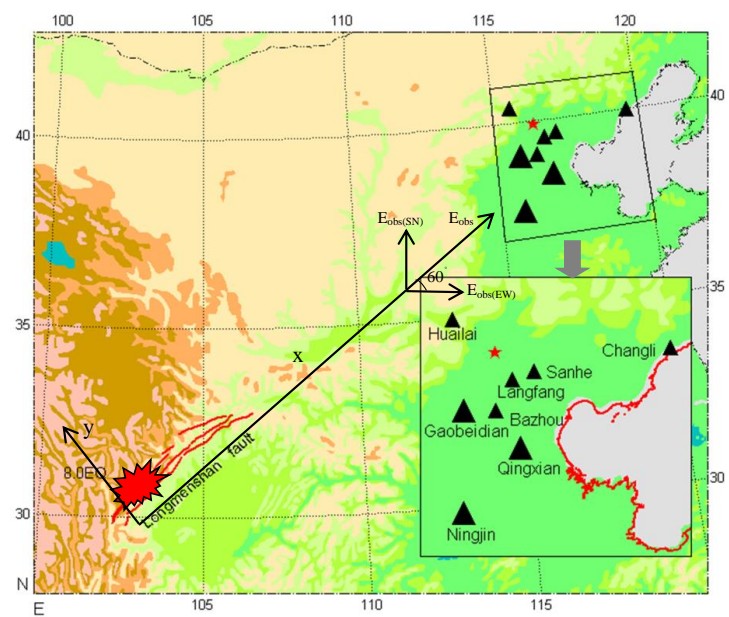

**Fig.3.** Distribution of the Wenchuan earthquake epicenter and observation stations. Black solid
triangles present the related locations of observation stations in Hebei electromagnetic observation
network, bigger ones indicate the stations where abnormal information was recorded and the red
star denotes Beijing (Li et al., 2013, Fig.1). A ground surface coordinate system is added.



On the base of the work of rock experiments conducted under laboratory
conditions, there is a reason to believe that a giant seismo-telluric current is generated
when the main rupture took place during the Wenchuan earthquake on 12 May 2008
and that this current mainly propagates along the Longmenshan fault. At the same
time a strong electrical field induced by this current suddenly increased. This
electrical field was recorded at the ground-based Gaobeidian observing station, 1440
km away from the epicenter of the shock, with a SN maximum amplitude of 70 mm,
i.e. 1.3 mV m$^{-1}$ (Li et al., 2013), that is $E_{obs(SN)} = 1.3$ mV m$^{-1}$ in the following
statement (Fig. 3).
In order to establish a relationship between the seismo-telluric current during the
main event and the observable ground electrical signals registered at Gaobeidian
station, we consider that a finite length current dipole source, with the length being
the main rupture L=150 km of the Wenchuan earthquake and the current I, is
co-located with the Longmenshan main fault (x-direction), with the depth being d=19
km. Then one can refer to Fig.1 with ionospheric effect.
A coordinate system on the Earth's surface (see Fig.3) is set up to calculate the
observable electrical field along the x-direction $E_{obs}$ according to the electrical value
$E_{obs(SN)}$=1.3 mV m$^{-1}$ recorded at the Gaobeidian station. The Gaobeidian station lies in
the extended line of the Longmenshan fault, which trends northeast and dips about
60 ˚west (Xu, 2009). Other locations of stations are shown in Fig.1 of Li et al. (2013)
and here they are shown in Fig.3 which includes a ground surface coordinate system.
From Fig.3, we see that the electrical filed component intensity along the x-direction
is about $|E_x|$=$E_{obs}$=1.5 mV m$^{-1}$ ($E_{obs(SN)}$=sin60 °×$E_{obs}$=1.3 mV m$^{-1}$ → $E_{obs}$=1.5 mV
m$^{-1}$).
As the observing frequency of the electromagnetic observation system is 0.1-10
Hz and the recorder belongs to a real-time analog record, it is not easy to finger out
the right frequency of the signals registered at the Gaobeidian station during the
maximum stage prior to the Wenchuan earthquake. We set the main frequency f=1 Hz
during our calculations although the information is of a short period ~0.1-0.3 s and a
large amplitude ~1.3 mV m$^{-1}$ (Li et al., 2013) and frequency bands (0.4-3 s and
0.05-0.1 s) with various amplitudes were observed (Guan et al., 2003). At the same
time, the results of 2D MT inversion in the Longmenshan fault show that the apparent
resistivity logarithm is ～1-4.8 (Zhu et al., 2008) and it is a wide range.

















**Fig.4.** The calculated value of |Ex|, expected at the observation location (1,440 km, 0, 0) due to a
dipole source of L=150 km, I=1 A, f=1 Hz and d=19 km (Fig.1), as a function of the typical
crustal materials conductivity σ both in Earth-air model (red line) and in Earth-air-ionosphere
model (blue dot line).

Fig. 4 shows the calculated values of |Ex|, expected at the observation location
(1,440 km, 0, 0) due to a dipole source of L=150 km, I=1 A and d=19 km (Fig.1), as a
function of the typical crustal materials conductivity σ. Comparing with the red line
with the blue dot one, the ionospheric effect is clearly displayed throughout the
variation of the crustal conductivity. A rapid attenuation (exceed 20 orders) of the
field values indicates the importance of the conductivity σ. It is difficult to specify the
average conductivity σ (referred to as $\sigma_1$ in the context) of the homogeneous Earth
medium, even for the typical Wenchuan area. However, combined with f=1 Hz here,
the skin-depth depends on the conductivity σ, given by the formula $\delta = (\pi f \mu_0 \sigma)^{-\frac{1}{2}}$.
Taken the depth d=19 km into account, here $\delta = d = 19$ km and the calculated $\sigma_1$
is attained, i.e. $\sigma_1 = 7.0 \times 10^{-4}\,\mathrm{S\,m^{-1}}$, which is advantageous to radiate
electromagnetic waves within this depth.
Using the same parameters as above, the simulation results show that the
seismo-telluric current along the main fault needed to produce an electrical ground
signal $E_{obs(SN)}$ = 1.3 mV m$^{-1}$ at the Gaobeidian station when the Wenchuan event
occurred, is about $5.3 \times 10^4$ kA with the ionospheric effect and $3.7 \times 10^5$ kA without the
ionospheric effect. As it is expected, these two results have one order (×10)



difference from each other. While the former is more reasonable under this conditions
because the seismo-telluric current produced by the Wenchuan main rupture is
specified.
**4.2. Detectability under the ionospheric effect**
Now according to the Wenchuan earthquake example, the seismo-telluric current
source (f=1 Hz, d =19 km, L=150 km, and a current I=$5.3 \times 10^4$ kA considering the
Earth-air-ionosphere model) is thought of as a powerful finite length dipole source.
Fig.5 displays the fluctuations of the surface electrical fields with and without
ionospheric effect for the Wenchuan source along x-axial direction. It shows no
obvious ionospheric effect within 300 km, while this effect is roughly up to 1
magnitude from ~800 km. The gap becomes larger as the distance extends, 2
magnitudes from ~4000 km, and then it keeps this gap till 10,000 km. Under this
condition, considering the observable signal 1.5 mV m$^{-1}$ at Gaobeidian station before
the Wenchuan epicenter, the distance recorded such a signal must be ～1500 km
(blue arrow) with ionospheric effect, or it is only ～800 km (red arrow) without
ionospheric effect. So the ionosphere facilitates the electromagnetic wave propagation,
as if the detectability of the system is improved effectively and it is easier to record a
signal even for stations located at distances beyond their detectability threshold.

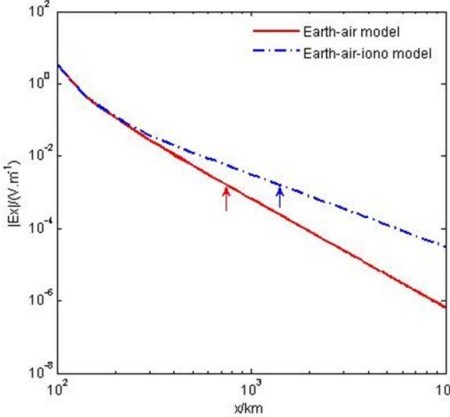

**Fig.5.** The Wenchuan source producing electric field |Ex| decay curves as a function of the
distance along x-axial direction with ionospheric effect (blue dot line), as well as without
ionospheric effect (red line). The electric field |Ex|=1.5 mV m$^{-1}$ is labeled by a red arrow and a
blue one respectively.



**5 Discussion**

In very recent years, there is an increasing amount of evidence that during some last stages of the long term process of preparation, there could be a transfer of energy between lithosphere and the above layers of atmosphere and ionosphere, so as to introduce the concept of a lithosphere–atmosphere–ionosphere coupling (LAIC) among the three involved layers of the Earth system (Pulinets et al., 2000; Hayakawa and Molchanov, 2002; Molchanov et al., 2004; Pulinets and Ouzounov, 2011). On one hand, the 'energy source' is usually thought to be beneath the Earth and related with tectonic activities in the lithosphere. On the other hand, numerous rock-pressure experiments and electromagnetic observations associated with seismic activities have already proved that a giant electrical current and an abrupt increase of electromagnetic signals are motivated when the main rupture occurs for stressed-rocks. These phenomena happed on May 9 2008, 3 days before the Wenchuan event, which hypocenter lies in mid-crust. The strong seismo-telluric current is thought to run mainly along the Longmenshan fault and electromagnetic oscillations, induced by the current and predominated by ULF frequency band, propagate up to ionosphere and give rise to perturbations of ionospheric parameters. Some of these parameters have been investigated, such as GPS TEC and f0F2 (Yu et al.,2009; Xu et al., 2010; Akhoondzadeh et al., 2010), DEMETER satellite O+ density (Zhang et al., 2009), electron density and electron temperature (Zeng et al., 2009), and so on. Fortunately, all these study results present a climax on May 9 and this indicates a lithosphere–atmosphere–ionosphere coupling or interaction aroused by these electromagnetic signals prior to the Wenchuan event.

Unfortunately, at present, most of investigations put emphases on the effect of earthquakes upon the ionosphere and few of them pay attention to an inverse problem, that is the ionospheric influence on the electromagnetic waves passed through.

The ionosphere, as a part of the electrical conducting region of the upper atmosphere, can enhance electromagnetic fields and make the decay slow when an observation is within ionospheric range and the ionosperic effect can be up to 1-2 magnitudes of the electrical fields in our simply three-layer model for some specified parameters we select.

Considering the Wenchuan event, the electrical signals from the lithosphere interact with the ionosphere and are improved simultaneously, and then registered at 1440km Gaobeidian station with the amplitude of 1.3 mV m$^{-1}$. This electrical field is



used to simulate the seismo-telluric current produced by the Wenchuan main rupture
in an Earth-air-ionosphere model together with an Earth-air model. The results present
that, the seismo-telluric currents with and without ionospheric effect must be about
$5.3 \times 10^{4}$ kA and $3.7 \times 10^{5}$ kA respectively. Compared with the expected seismo-telluric
current ~10–100 kA of the "Alum Rock" $M_{W}$=5.6 earthquake for an observed 30 nT
pulse at 1 Hz and D=2 km (Bortnik et al., 2010), this result is probably in a reasonable
range.
However, firstly, the total rupture of the Longmenshan fault concerned with the
Wenchuan main shock is extremely complicated that contains tenths rupture stages
and several pauses, totaled 90 s for the whole rupture process (~300 km), according
to Zhang et al.,(2009). Thus the total surface rupture ~300 km is nevertheless not
used here. While performing the analysis on only the primary 30 s, a main stage of the
Wenchuan earthquake, out of 90 s as we have selected L=150 km above, is expected
to be representative of the majority of the rupture to generate a seismo-telluric current.
Secondly, three medium are thought of as a homogeneous isotropic medium in our
models and with the same average conductivity value for each one, especially for the
wenchuan area. However, the Earth conductivity plays a so important role that can
predominately affect the fluctuations of the electrical fields as shown in Fig.4
although no one exactly knows the right conductivity of the bottom medium. The
value $\sigma_{1} = 7.0 \times 10^{-4}\,\mathrm{S\,m^{-1}}$ taken part in all analysis is estimated when the
observing frequency range f=0.1-10 Hz and the hypocenter depth d=19 km of the
Wenchuan main event are taken into account for the skin-depth formula. One must
also mention that we use f=1 Hz in our calculations because we cannot distinguish the
right frequency of the recorded analog signals. All these can underscore our
simulation results.
While these disadvantageous selections maybe are not so important at the same
time because the key point of this paper is of ionospheric influence on
electromagnetic waves and our investigation attains advantageous results.

**6 Conclusions**

In this paper, a three-layer (Earth-air-ionosphere) physical model, as well as a
two-layer (Earth-air) model, is employed to investigate the ionospheric effect on the
wave fields for a finite length dipole current source co-located with the main fault of
an earthquake when an observing location distance is up to one thousand kilometers





or even more. For a dipole source with specified parameters of the length L=150 km,
the current I=1 A, and the depth d=19 km, the results show that all fields are free of
the ionospheric effect for different frequencies in relative short ranges, e.g., ～600
km for f=0.1 Hz, which implies the ionospheric influence on electromagnetic field
transmissions can be neglected within this range. However, the ionosphere can
strengthen the field amplitude and make the decay slow when an observation is out of
this range and the ionosperic effect can be up to 1-2 magnitudes of the electrical
fields.

This is suitable for the 12 May 2008 Wenchuan $M_S$=8.0 earthquake during which

a strong electromagnetic signal with an amplitude of ~1.3 mV m$^{-1}$, is recorded by the
Gaobeidian ULF (f=0.1-10 Hz) observing station 1440 km away from the epicenter.
The main fault rupture producing a current is equivalent to a finite length dipole
current source, with a nucleation depth of 19 km and a length of 150 km. Considering
the Earth-air-ionosphere model, the expected current for the most typical properties of
Wenchuan area is of $5.3 \times 10^4$ kA, which is of one magnitude smaller than the current
value of $3.7 \times 10^5$ kA obtained with the Earth-air model free of ionospheric effect. On
the contrary, a signal introduced by a seismic activity can be advantageously recorded
by a remote station under the ionospheric effect as if the detectability of the system is
improved effectively.

*Acknowledgements and data.* The authors are grateful to the National Natural Science
Foundation of China and this work was sponsored by the project Simulation and
Interpretation of the Spatial Electromagnetic Phenomena Coupling before the
Wenchuan $M_S$8.0 Earthquake under grant agreement n ⁰41204057. The data presented
in this paper are available to the e-mail: limeixuxl@seis.ac.cn.

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
