# Peer review of "Ionospheric influence on the seismo-telluric current related to electromagnetic signals observed before the Wenchuan $M_S$=8.0 earthquake"

_Solid Earth, 2016_

## Referee Comment (RC1) · Dr. Freund (Referee) · 17 Jul 2016

I have read this paper with great interest. It is a good paper, addressing an important issue, namely the ability to record over long distances the ultralow frequency EM emission prior and during a major seismic event, here the M=8 Wenchuan earthquake of May 12, 2008. The paper focuses on the enhancement of the EM signal in the waveguide between the surface of the Earth and ionosphere.

I tried my best to help the authors improve the English of their text. see attached annotated pdf.

I have a few questions regarding (1) the introduction of "peroxy defects" in the early part of the paper, and (2) geometry of the actual situation on the ground, i.e. the SW-NE trending Longchanming fault and the location of the receiving stations of the emitted ULF waves in the NE direction, almost exactly in the extended direction of the fault.

(1) Since I am the one who recognized for the first time the presence of "peroxy defects" and their ubiquity in essentially all crustal rocks down to at least 35-45 km depth, I am sensitive to the way "peroxy defects" are introduced. Here they are introduced without direct reference. This reference is given only later to my papers cited such as Freund, F.: Charge generation and propagation in igneous rocks, J. Geodynamics, 33, 543–570, 2002., and Freund, F.: Toward a unified solid state theory for pre-earthquake signals, Acta Geophys., 58(5), 719–766, 2010. It stands to reason to insert those citations where the word "peroxy defect" is first used.

I also want to point out that, while it is correct that I first developed the idea that stresses will generate dislocations which then activate peroxy defects and release mobile positive hole charge carriers, I have since come to realize that this is an unlikely mechanism. Much more probably is that stresses cause ever so slight displacements of mineral grains in the rocks, which in turn lead to the activation of peroxy defects that preferentially sit on or across grain boundaries. A more recent reference relevant to this question is:

Scoville, J., J. Sornette and F. T. Freund (2015). "Paradox of peroxy defects and positive holes in rocks Part II: Outflow of electric currents from stressed rocks." Journal of Asian Earth Sciences 114, Part 2: 338-351.

(2) The authors do not discuss one important aspect of their model, namely the fact that a linear dipole will emit more intensity perpendicular to the dipole axis and, theoretically, zero intensity in the direction of the dipole axis. However, the location of the receiving stations from which the authors obtain their data is nearly exactly in the direction of the dipole axis. The question arises: how much stronger would the received EM signals have been, if data had been used from receiver stations at about right angle to the length of the Longshanming fault? Though I don't think that this omission is a major weakness of this paper, it would be advisable that the authors include a paragraph to indicate that this question exists and the numerical results may be different.

Please also note the supplement to this comment:
http://www.solid-earth-discuss.net/se-2016-89/se-2016-89-RC1-supplement.pdf

**Supplement:**

[revised manuscript text omitted]

Something is wrong here: according to Figure 1 the Earth medium is at z<0 and the air medium is at z>0.

As a comparison, a two-layer model (Earth-air model) including in Earth
medium (z > 0), as well as air medium (z < 0), is also established during the study.
All the corresponding parameters described are the same as these of
Earth-air-ionosphere model.

We assume that the total space is non-magnetic and that the magnetic
permeability μ variations are negligible in the different layers, i.e. $\mu_1 = \mu_0 = \mu_{-1} =$
$4\pi \times 10^{-7}$ Farad m$^{-1}$. However, the ionosphere as the electrically conducting
section of the upper atmosphere  important role  the electromagnetic
propagation that we set $\varepsilon_{-1} = 5\varepsilon_0$ when an ionospheric effect on electromagnetic
transmission is taken into consideration. On the same manner we have $\varepsilon_1 = \varepsilon_0 =$
$8.854 \times 10^{-12}$ Farad m$^{-1}$, i.e. $\varepsilon_1$ is not considered as zero during all calculations.
Under these conditions, the formula listed above are still suitable and more
explanations about the potential formulation of a horizontal electric dipole can be
found in the Appendix A of Key (2009) and related programs are available with an
access to the website (http://marineemlab.ucsd.edu/). The horizontal finite length
dipole source can be  as  integral of an infinite small horizontal dipole during
related calculations.

**3 Simulation results**

According to these two models  above, several free parameters must be
specified in order to investigate the attenuation characteristics of the electromagnetic
fields emitted by a long x-directed dipole current source. As for the parameters of the
dipole current source, we select L=150 km, the Wenchuan earthquake main rupture
stage within 30 s out of 90 s (∼300 km) based on Zhang et al., (2009, Fig.1), the
depth d =19 km (Xu, 2009), the hypocenter depth of the Wenchuan case and the
current is set to be I=1 A temporarily. Here, the Earth is considered to be an isotropic
media with an average conductivity $\sigma_1$, and we assume $\sigma_1 = 1.0 \times 10^{-3}$ S m$^{-1}$ at
this time, i.e. $\rho_1 = 10^3$ ohm · m, although the ground conductivity depends not only
on the local petrology, but also on the porosity, temperature, and pressure (e.g., Wait,
1966). All these parameters are common to two models.  parameter
$\varepsilon_{-1} = 5\varepsilon_0$ is of most importance during the calculation in three-layer model in that it
potentially can affect the transmission of electromagnetic waves produced by the
dipole beneath the Earth, and possibly  the Earth-atmosphere-ionosphere
electromagnetic coupling.

[Figure]

[Figure]

[Figure]

**Fig.2.** Electric field $|E_x|$ decay curves along x-axial direction as a function of the observing distance for the Cartesian coordinate system with different frequencies. Red solid lines stand for electric field curves for Earth-air model and blue dot lines denote electric field curves with the ionospheric effect for Earth-air-ionosphere model.

(a) Total $|E_x|$ for f=0.01 Hz;    (b) Total $|E_x|$ for f=0.1 Hz; (c) Total $|E_x|$ for f=1 Hz;    (d) Total $|E_x|$ for f=10 Hz;

Fig.2a-d displays electric field component amplitude $|E_x|$ decay curves along the x-axial direction with the frequencies f=0.01 Hz, f=0.1 Hz, f=1 Hz, and f=10 Hz respectively for the Cartesian coordinate system up to ～10,000 km on the Earth's surface.

It can be seen from Fig.2a-d, first, the electrical field with "high" frequency has a big attenuation although all curves for both Earth-air model (red solid lines) and Earth-air-ionosphere model (blue dot lines) decay rapidly as the distance increases. Second, each group of curves run at the same level for one fixed frequency, e.g., f=1 Hz, when an observing point is located at a relative near distance, ～300 km for f=1 Hz (Fig.2c) for example. That is to say, the ionospheric influence on electromagnetic field transmissions can be neglected within this range. However this range changes

for different frequencies and it becomes smaller as the operating frequency of the current source increases (e.g., more than 1000 km for f=0.01 Hz (Fig.2a) and only

~200 km for f=10 Hz (Fig.2d)). Third, the most important result is, as the distance increases, field curves with an ionospheric effect (blue dot lines) run  a different from that of curves without an ionospheric effect (red solid lines) and the ionospheric lines attenuate more slowly. Now, this kind of ionospheric influence be neglected . The ionospheric difference is about 1

magnitude (×10) for all the frequencies listed and even once up to 2 magnitudes for f=10 Hz within the range shown in Fig.2. For example, the ionospheric difference value shows 1 magnitude from ～840 km, up to 2 magnitudes from ～3,700 km for f=10 Hz (Fig.2d).

**4 The Wenchuan $M_S$=8.0 earthquake as a sample**

**4.1 Estimating the seismo-telluric current magnitude**

[Figure]

with the geometry shown the intensity of the EM emission perpendicular to the Longmengshan fault should be much higher as indicated in the graphics pasted into your figure. Any comment?

[revised manuscript text omitted]

---

## Author Comment (AC1) · 20 Jul 2016

[revised manuscript text omitted]

Recently, the work of Freund et al. (Freund and Wengeler, 1982, Freund, 2002, 2009, 2010; Freund and Sornette, 2007; Scoville et al., 2015) has gained a new insight into the production of current and electromagnetic signals in stressed rocks. As rocks upon stressing, stresses cause slight displacements of mineral grains in the rocks, which in turn lead to the activation of peroxy defects that preferentially sit on or across grain boundaries. The peroxy break-up leads to positive holes $h^*$ and the $h^*$ are able to flow from stressed to unstressed rock, traveling fast and far by way of a phonon-assisted electron hopping mechanism using energy levels at the upper edge of the valence band. A gabbro sample($30 \times 15 \times 10$ cm$^3$)from Shanxi, China, was used in 
[revised manuscript text omitted]
 were improved effectively and it would be easier to record a signal even at stations located beyond their detectability threshold.

[Figure]

**Fig.5.** The Wenchuan source producing electric field |Ex| decay curves as a function of the distance along x-axial direction with ionospheric effect (blue dot line), as well as without ionospheric effect (red line). The electric field |Ex|=1.5 mV m$^{-1}$ is labeled by a red arrow and a blue one respectively.

**4.3. Wave 2-D distribution**

We perform electromagnetic wave fields for the Wenchuan source and this is done in the ground plane region -1,000 km<x<1,000 km and -1,000 km<y<1,000 km in order to visualize the 2-D distribution of the wave power surrounding the electrical source.

Figure 6 displays the 2-D power distributions of the electrical field components |Ex|, |Ey| and the total |E| ($|E|^2=|Ex|^2+|Ey|^2$) after making a logarithm calculation on the Earth's surface. It can be seen firstly from Figure 6a that there is an obvious constant strong power along the current element length (-75 km<x<75 km) in the x-direction. The electrical value in this area is not discussed here because it is usually considered not precise. Then the strong field radiates outward surrounding four main axes, indicating 1 order rough decay of the field at ～160 km, 2 orders of magnitude at ～320 km from the source endpoint in the x-direction. There is only 3 orders decay till 1,000 km away because of the ionospheric facilitating effect on the field and it keeps a strong value (～1.86 mV) which can be fairly recorded by the stations. However, there are also weak power areas along lines, which form 45 °angle with the principal axis for the electrical field power |Ex| (Figure 6a). Complementally, the electrical field power |Ey| (Figure 6b) is basically characterized by strong power areas between two main axes, as well as weak ones along four chief axes. The power distribution of the total |E| consequently presents to be symmetry to the center circle outside of the source (Figure 6c), which also indicates that the radiating patterns of the electrical field power |Ex| and the electrical field power |Ey| are complementary (One is strong area and the other is weak area) each other surrounding the source.

[Figure]

(a)                                    (b)

(c)

**Fig.6.** 2-D distributions of electrical field power |Ex| (a), |Ey| (b) and total |E| (c) after a logarithm calculation for the Wenchuan source using Earth-air-ionosphere model.

**5 Discussion**

In very recent years, there is an increasing amount of evidence that during some last stages of the long term process of preparation, there could be a transfer of energy between lithosphere and the above layers of atmosphere and ionosphere, so as to introduce the concept of a lithosphere–atmosphere–ionosphere coupling (LAIC) among the three involved layers of the Earth system (Pulinets et al., 2000; Hayakawa and Molchanov, 2002; Molchanov et al., 2004; Pulinets and Ouzounov, 2011). On one hand, the 'energy source' is usually thought to be beneath the Earth's surface and related to tectonic activities in the lithosphere. On the other hand, numerous rock-pressure experiments and electromagnetic observations associated with seismic activities have already proved that a giant electrical current and an abrupt increase of electromagnetic signals occur during the main rupture of stressed-rocks. These phenomena happed on May 9 2008, 3 days before the Wenchuan event, which hypocenter lies in mid-crust. The strong seismo-telluric current is thought to run mainly along the Longmenshan fault and electromagnetic oscillations, induced by the current and predominated by ULF frequency band, propagate up to ionosphere and give rise to perturbations of ionospheric parameters. Some of these parameters have been investigated, such as GPS TEC and f0F2 (Yu et al.,2009; Xu et al., 2010;

Akhoondzadeh et al., 2010), DEMETER satellite O+ density (Zhang et al., 2009), electron density and electron temperature (Zeng et al., 2009), and so on. Fortunately, all these study results present a climax on May 9 and this indicates a lithosphere–

atmosphere–ionosphere coupling or interaction aroused by these electromagnetic signals prior to the Wenchuan event.

Unfortunately, at present, most of investigations put emphases on the effect of earthquakes upon the ionosphere and few of them pay attention to an inverse problem, that is the ionospheric influence on the electromagnetic waves passing through.

The ionosphere, as a part of the electrical conducting region of the upper atmosphere, can enhance electromagnetic fields and cause the decay as a function of distance to slow down when an observation is within ionospheric range and the ionosperic effect can be up to 1-2 magnitudes of the electrical fields in our simply three-layer model for some specified parameters we have selected here.

Considering the Wenchuan event, the electrical signals from the lithosphere interact with the ionosphere and are at the same time enhanced, and then registered at

1440km Gaobeidian station with the amplitude of 1.3 mV m$^{-1}$. This electrical field is used to simulate the seismo-telluric current produced by the Wenchuan main rupture in an Earth-air-ionosphere model together with an Earth-air model. The results present that, the seismo-telluric currents with and without ionospheric effect must be about

$5.3 \times 10^7$ A and $3.7 \times 10^8$ A respectively. Compared with the expected seismo-telluric current ~10–100 kA of the "Alum Rock" $M_W$=5.6 earthquake for an observed 30 nT

pulse at 1 Hz and D=2 km (Bortnik et al., 2010), this result is probably in a reasonable range.

 However, firstly, the total rupture of the Longmenshan fault during the
Wenchuan main shock is extremely complicated that comprises of tenths of rupture
stages and several pauses, totaling 90 s for the whole rupture process ($\sim$300 km),
according to Zhang et al.,(2009). Thus the total surface rupture $\sim$300 km is
nevertheless not used here. While performing the analysis on only the primary 30 s, a
main stage of the Wenchuan earthquake, out of 90 s as we have selected L=150 km
above, is expected to be representative of the majority of the rupture to generate a
seismo-telluric current. Secondly, three medium are thought of as a homogeneous
isotropic medium in our models and with the same average conductivity value for
each one, especially for the wenchuan area. However, the Earth conductivity plays
such an important role that it predominately affects the fluctuations of the electrical
fields as shown in Fig.4 although no one exactly knows the right conductivity of the
Earth medium at the rupture depth. The value $\sigma_1 = 7.0 \times 10^{-4}$ S m$^{-1}$ taken part in
all analysis is estimated when the observing frequency range f=0.1-10 Hz and the
hypocenter depth d=19 km of the Wenchuan main event are taken into account for the
skin-depth formula. One must also mention that we use f=1 Hz in our calculations
because we cannot identify the actual frequencies in the recorded analog signals. All
these can underscore our simulation results.

 While these disadvantageous selections maybe are not so important at the same
time because the key point of this paper is of the ionospheric influence on
electromagnetic wave propagation and our investigation attains advantageous results.

 The "selectivity" or "orientation" of the electromagnetic information is a very
important character during seismic activities (Varotsos and Lazaridou, 1991). For a
finite length dipole source of the Wenchuan earthquake, its 2-D distributions of
electrical field component |Ex| and |Ey| , which are orthogonal each other, on the
Earth's surface shows there are strong field power areas and weak field power areas
around the source as illustrated by [Bortnik et al., 2010]. While the radiating pattern
of the total |E| in this investigation is symmetry to the center circle outside of the
source which indicates a signal is always registered to anyone direction if a system is
designed to measure the total field |E| or both of |Ex| and |Ey| components instead of
only one. This result also basically supports the practices of "selectivity" or
"orientation", the observing reality before the Wenchuan earthquake described by Li
et al.[2013], for example, 'Compared with the EW (East-West) orientation, the
electromagnetic signal is more obvious in the SN (South-North) orientation'. The selectivity effect is a complex phenomenon that may be attributed to a superposition of the following three factors: "source characteristics", "travel path" and

"inhomogeneities close to the station" [Varotsos and Lazaridou, 1991; Varotsos et al.,

2005]. Analytical solutions of Maxwell equations [Varotsos et al., 2000], as well as numerical ones [Sarlis et al., 1999], convince that selectivity results from the fact that earthquakes occur by slip on faults which are appreciably more conductive than the surrounding medium.

**6 Conclusions**

[revised manuscript text omitted]

Xu, C., Di, Q. Y., Fu, C. M. and Wang, M. Y.: The contrast of response characteristics between large power long dipole and circle source, Chinese J.

Geophys, 55(6), 2097–2104, doi: 10. 6038/ j. issn. 0001–5733. 2012. 06. 03, 2012, (in Chinese with English abstract).

Xu, T., Hu, Y., Wu, J., Wu, Z., Suo, Y., and Feng, J.: Giant disturbance in the ionospheric F2 region prior to the $M$8.0 Wenchuan earthquake on 12 May 2008, Ann. Geophys., 28, 1533–1538, 2010.

Xu, X. W.:Album of 5.12 Wenchuan 8.0 earthquake surface ruptures. Seismological press, 2009 (in Chinese with English abstract).

Yamauchi, T., Maekawa, S., Horie, T., Hayakawa, M., and Soloviev, O.: Subionospheric VLF/LF monitoring of ionospheric perturbations for the 2004 Mid-Niigata earthquake and their structure and dynamics, J. Atmos. Sol. Terr. Phys., 69, 793–802, 2007.

Yu, T., Mao, T., Wang, Y. G., and Wang, J. S.: Study of the ionospheric anomaly before the Wenchuan earthquake, Chinese Science Bulletin, 54(6): 1086–1092, doi: 10.1007/s11434-008-0587-8, 2009 (in Chinese with English abstract).

Zeng, Z. C., Zhang, B., Fang, G. Y., Wang, D. F., and Yin, H. J.: The analysis of ionospheric variations before Wenchuan earthquake with DEMETER data, Chinese J. Geophys., 52(1): 11–19, 2009 (in Chinese with English abstract).

Zhang, X., Shen, X., Liu, J., Ouyang, X., Qian, J., and Zhao, S.: Analysis of ionospheric plasma perturbations before Wenchuan earthquake. Nat. Hazards Earth Syst. Sci., 9: 1259–1266, 2009.

Zhang ,Y., Feng, W. P., Xu, L. S., Zhou, C. H., and Chen, Y. T.: Spatio-temporal rupture process of the 2008 great Wenchuan earthquake, Science in China Series D: Earth Sciences, 52 (2), 145–154, 2009.

Zhu, Y. T., Wang, X. B., Yu, N., Gao, S. Q., Li, K., and Shi, Y. J.: Longmenshan magnetotelluric deep structure and the Wenchuan earthquake ($M_S$8.0), Acta Geologica Sinica, 82 (12), 1769–777, 2008 (in Chinese with English abstract).

---

## Referee Comment (RC2) · Anonymous Referee #2 · 25 Jul 2016

The manuscript presents two and three layer models studies and the authors conclude that upto 300 km the signal is affected by the ionosphere, whereas ionosphere influence the signal if it is measured more than 300 km. It is not very clear how the authors have tried to link with the Wenchuan earthquake, except that some of investigators have observed ionospheric perturbations in the ionosphere. If the focus of the earthquake is 19 km, how consideration of homogeneous isotropic conductivity values can be justified?. The authors findings that ionosphere influence em propagation beyond 300 km is a trivial exercise, ofcourse it all depends on frequency and conductivity of medium. But with such conclusion, one may not relate with the Wenchaun earthquake. Numerous papers have been published on Wenchuan earthquake,

several authors have discussed surface, subsurface, atmosphere and ionospheric parameters, If the authors have some observed MT data or ground data, it would be better to show anomalies associated with the Wenchuan earthquake. The authors have not mentioned the relevance of their findings with Wenchaun earthquake in the abstract, but in discussion they made an effort to related with the Wenchuan earthquake. The manuscript is full of English language and grammar problems.

---

## Author Comment (AC2) · 27 Jul 2016

We thank the reviewer for his comments! Answers are given below.

Background of the investigation in this paper

There are an increasing number of reports that ULF electromagnetic emissions are recorded at several, hundred, and even several thousand kilometers away from the epicenters before some strong earthquakes. It is also well established that, during rock experiments conducted under laboratory conditions, a strong electrical current and electromagnetic emissions are produced when rocks are stressed, especially at the stage of the main rupture although there is no clear physical mechanisms of these

electromagnetic emissions.

As the development of satellite Earth Observation (EO), there is an increasing amount of evidence that during some last stages of the long term process of preparation, there could be a transfer of energy between lithosphere, atmosphere and ionosphere, so as to introduce the concept of a lithosphere–atmosphere–ionosphere coupling (LAIC) or interacting among the three involved layers of the Earth. Several tentative LAIC models have been constructed based on ground-based and ionospheric observations prior to strong earthquakes and the investigation of influence of earthquake related external electrical field on ionospheric parameters has been gained much achievement. At the same time, the ionosphere plays an important role in electromagnetic propagation at Extremely Low Frequency (ELF) and Very Low Frequency (VLF), the ground and the ionosphere are good electrical conductors and form a spherical Earth-ionosphere waveguide. In addition, in the Controlled Source Electromagnetic (CSEM) method, widely used in petroleum exploration or mining, the ionospheric influence on electromagnetic (EM) fields should be considered when the distance between a large-scale and large-power fixed source and the receiver is up to one thousand kilometers. EM fields can be amplified in the ionosphere as it is shown when we use analytical solutions of Maxwell equations, as well as numerical ones of the "Earth-ionosphere" mode with a source on the Earth's surface or in the subsurface of the Earth.

Ground-based ULF electrical emissions during the Wenchuan MS=8.0 earthquake

The Hebei ULF (0.1-10 Hz) electromagnetic observation network was constructed at the beginning of 1980s after the occurrence of the July 28, 1976, Tangshan MS 7.8 EQ with the aim of monitoring fluctuations in the electromagnetic radiations before seismic activities mainly around Beijing. More details of the observation system can be found in Zhuang et al. (2005) and Li et al (2013).

The system measures electrical signals and a DJ-1 recorder is employed to record the potential difference between two electrodes (SN, South-North and EW, East-West).

The recording method uses an analog automatic real-time continuous pen recorder with a speed of 1 mm/s. In general, only parallel lines with perpendicular automatic clock marked signals on the record paper around a drum and six lines are left per hour. A blank record paper replaces the recorded one at 9:00 AM (local time) everyday (seen attached Fig.1). Attached Fig.1 is copy of a part of normal original record (EW component) from 9:00AM, 1 to 9:00 AM, 2, May, 2008 at Gaobeidian station. Corresponding letters: A denotes a start record point, i.e. 9:00 AM, 1, May; B is a normal record line; C indicates a manually marked time, i.e. 20:00, 1, May; D indicates an automatic marked perpendicular hour line, i.e. 20:00, 1, May; and E shows an end record point, i.e. 9:00 AM, 2, May.

During the period from January 2007 to December 2008, electrical emissions were recorded at three among four (only four stations run normally during this time) stations (Fig.3 in revised paper attached) and the recording at Gaobeidian station shows a typical fluctuation character. Anomalous emissions first appeared at the end of October 2007 and the information was not recorded everyday but it is mainly accumulated in SN direction.

On 2 November 2007, our work team went to Gaobeidian station to check observing environment and eliminate probable interferences. Attached Fig.2 is a picture taken on 2 November 2007 at Gaobeidian station. The work team check real-time recording paper. This kind of situation lasted till the beginning of April, 2008, from when relative high frequency and large amplitude signals were recorded almost every day with a persistent time. On May 9, 2008, 3 days before the Wenchuan MS 8.0 EQ, the amplitudes of signals were suddenly subjected to an abrupt enhancement at the same time, between 5:00 AM and 7:00 AM, both in the SN and EW directions and the abnormity reached to the climax stage ($\sim$1.3 mV/m for electric field) till on 17 May, 2008. Attached Fig.3 is a picture taken on 7 May 2008 at Gaobeidian station. The work team check real-time recording paper.

During this period, the work team went to most of stations to check related recordings and their observing environment. And they were right at Sanhe station when the Wenchuan earthquake took place. Attached Fig.4 is a picture taken on 7 May 2008 at Ningjin station. One of the observers of Langfang station measured the magnitude of the recorded signals.

After May 18, the total signal amount decreases sharply and the character of the signals at this stage is more like that before April 2008. The SN information lasted till the end of September 2008 except for high emissions appearing before several powerful aftershocks. It is the first time that the abnormality is with such a large amplitude and such a long duration in the observation history of this network although several strong EQs were recorded before (Li et al., 2013).

Fortunately, our reseach team traced this kind of obvious emissions during all this period and went to the stations several times to check observing environment and eliminate probable interferences but found none. While the large Wenchuan MS=8.0 earthquake took place during this period. So this obvious ULF eminssions probably are related to this event.

As for the homogeneous isotropic conductivity of the Earth

The hypocenter depth of the Wenchuan main shock is 19 km and as a matter of fact, no one exactly knows the right conductivity of the Earth on one hand. On the other hand, it is difficult to the right frequency of the signals observed during the climax stage because of an analog observation and real-time signals are added together. So according to ULF observing frequency 0-10 Hz band, we set f=1 Hz during calculations. Then the advantageous conductivity of the Earth at this frequency is attained by the skin-depth formula. All these uncertain values maybe underscores our results, which is discussed in the paper.

As for English language and grammar problems appeared in the paper

We feel grateful to the reviewer for pointing out this problem. We have already corrected

some language and grammar problems according to Professor Dr. Freund's advices and we all try our best to improve our writing English.

In addition, the revised paper will be attached and all modified are in red. Figures attached are some ULF real-time recordings at Gaobeidian station: Fig.A Picture of real-time recordings from 9:00 AM, 13 to 9:00 AM, 14, February, 2008 at Gaobeidian station. Fig.B Picture of real-time recordings from 9:00 AM, 8 to 9:00 AM, 9, May, 2008 at Gaobeidian station. Fig.C Picture of real-time recordings from 9:00 AM, 12 to 9:00 AM, 13, May, 2008 at Gaobeidian station. Fig.D Picture of real-time recordings from 9:00 AM, 20 to 9:00 AM, 21, May, 2008 at Gaobeidian station.

Please also refer to:

Li, M., Lu, J., Parrot, M., Tan, H., and Zhang, X.: Review of unprecedented ULF electromagnetic anomalous emissions possibly related to the Wenchuan MS = 8.0 earthquake, on 12 May 2008. Nat. Hazards Earth Syst.Sci., 13(2), 279–286, 2013.

Zhuang J, Vere-Jones D, Guan H, et al. Preliminary Analysis of Observations on the Ultra-Low Frequency Electric Field in the Beijing Region. Pure & Applied Geophysics, 162(6), 1367-1396, 2005.

Please also note the supplement to this comment:
http://www.solid-earth-discuss.net/se-2016-89/se-2016-89-AC2-supplement.pdf

[Figure]

[Figure]

**Fig. 1.**
The first author of this paper

**Fig. 2.**

The first author of this paper

The analog real-time recorder

**Fig. 3.**

**Fig. 4.**

[Figure]

**Fig. 5.**

[Figure]

**Fig. 6.**

[Figure]

**Fig. 7.**

[Figure]

**Fig. 8.**

**Supplement:**

**Ionospheric influence on the seismo-telluric current related to electromagnetic signals observed before the Wenchuan $M_S$=8.0 earthquake**

Mei Li[1,2], Handong Tan[2] and Meng Cao[2]

⑴ China Earthquake Networks Center, China Earthquake Administration, No.5, Sanlihe Nanhengjie, Xicheng District, 100045 Beijing, China.

⑵ China University of Geosciences, No.29, Xueyuan Road, Haidian District, 100083 Beijing, China.

Corresponding author: Handong Tan, China University of Geosciences, No.29, Xueyuan Road, Haidian District, 100083 Beijing, China. (thd@cugb.edu.cn)

**Abstract.** A three-layer (Earth-air-ionosphere) physical model, as well as a two-layer (Earth-air) model, is employed in this paper to investigate the ionospheric effect on the wave fields for a finite length dipole current source co-located at a hypocenter depth and along the main fault of an earthquake when the distance between the epicenter and an observing station is up to one thousand kilometers or even more. The results show that all electrical fields are free of ionospheric effect for different frequencies in a relative short range, e.g., ～300 km for f=1 Hz, implying the ionospheric influence on electromagnetic fields can be neglected within this range, which becomes smaller as the frequency increases. However, the ionosphere can give a constructive interference to the waves passed through and make them decay slowly when an observation is out of this range and the ionosperic effect can be up to 1-2 magnitudes of the electrical fields. For an ground-based observable 1.3 mV m$^{-1}$ electric signal at f=1 Hz at 1,440 km away from the Wenchuan $M_S$=8.0 earthquake, the expected seismo-telluric current magnitude for the Earth-air-ionosphere model is of $5.0 \times 10^7 \, A$, one magnitude smaller than the current value of $3.7 \times 10^8 \, A$ obtained by the Earth-air model free of ionospheric effect. This indicates that the ionosphere facilitates the electromagnetic wave propagation, as if the detectability of the system is improved effectively and it is easier to record a signal even for stations located at distances beyond their detectability threshold. Furthermore, the radiating patterns of the electrical field components |Ex| and |Ey| are complementary each other although anyone 2-D power distribution of them shows strong power areas as well as weak ones, which is advantageous to register a signal if the observing system is designed to measure both of them instead of only one.

**Keywords.** Ionospheric influence on electromagnetic waves; The Wenchuan earthquake; Seismo-telluric current; 2-D power distribution

**1 Introduction**

The fact that Electro-Magnetic (EM) emissions accompany every stage of large earthquake preparations seems undebatable although short-term earthquake prediction is still one of the most challenging targets in Earth science today (Eftaxias et al., (2002). Meanwhile, the Ultra-Low Frequency (ULF) band is of particular interest because only EM signals in the ULF range and at lower frequencies originated in the Earth's crust can be easily recorded at the Earth's surface without significant attenuation comparing with 'high' frequency emissions that might be emitted at epicenter depths at more than 10 km, even several hundreds of kilometers. Recently, an increasing number of ground-based observing ULF electromagnetic emissions related to strong earthquakes have been recorded at a distance of several, hundred, and even several thousand kilometers. Some notable examples include the Loma Prieta $M$s=7.1 earthquake on October 17, 1989 (f=0.01-10 Hz, D=7 km, A=1.5 nT) (Fraser-Smith et al., 1990; Bernardi et al., 1991), as well as the Spitak $M_S$=6.9 earthquake on December 7, 1988 (f=0.005-1 Hz, D=200 km, A=0.2 nT) (Molchanov et al., 1992; Kopytenko et al., 1993). In addition, the geo-electric potential enhancement appeared 1–19 days before five of all six EQs with magnitude >5 that occurred within 75 km in Japan and its duration and intensity were several minutes to 1 h with an amplitude of 0.01–0.02 mV m$^{-1}$ (Uyeda et al., 2000). Qian *et al*. (2002) have reported the observation of ULF signals generated from Jiji earthquake of 21 September 1999 in Taiwan and recorded at many stations at distances of 300–900 km in South East China. Similarly, Ohta *et al*. (2002) have reported the observation of ULF/ELF emissions generated from Taiwan earthquake of 21 September, 1999 and recorded at Nakatsugawa station in Japan at a distance of up to 2000 km.

A more notable example reported by Li et al. (2013) is the Wenchuan $M$s=8.0 earthquake on May 12, 2008, a typical mid-crust, which resulted in great devastation and 69,000 deaths. This earthquake was preceded by more than one month of increasing anomalous ULF emissions with a climax starting on May 9, three days before the Wenchuan main shock (f=0.1-10 Hz, D=1,440 km, A=1.3 mV m-1).

Many simulating rock-pressure experiments were carried out in order to
understand the producing mechanism of the electromagnetic information associated
with seismic activities. Laboratory experiments by *Qian et al.*, [1996; 2003] and *Hao*
*et al.* [2003] present that, electromagnetic signals are always recorded when rock
samples are subjected to dynamic stresses. Electromagnetic pulses of shorter-period
appearing at the last stage of the experiment may be induced by instantaneous electric
current of the accumulated charge during the stress acceleration.The work of Freund
et al. (Freund and Wengeler, 1982, Freund, 2002, 2009, 2010; Freund and Sornette,
2007; Scoville et al., 2015) has gained a new insight into the production of current
and electromagnetic signals in stressed rocks. As rocks upon stressing, stresses cause
slight displacements of mineral grains in the rocks, which in turn lead to the activation
of peroxy defects that preferentially sit on or across grain boundaries. The peroxy
break-up leads to positive holes $h^*$ and the $h^*$are able to flow from stressed to
unstressed rock, traveling fast and far by way of a phonon-assisted electron hopping
mechanism using energy levels at the upper edge of the valence band. A gabbro
sample($30{\times}15{\times}10$ cm$^3$)from Shanxi, China, was used in the test and a 55 nA current
recorded about 2 seconds before failure, with the load being at about 30,000 lbs and
the maximum spike reaches 450 nA when the main failure took place (Freund, 2009).

Up to now, no clear explanation has been given although several physical
mechanisms have been proposed to interpret the generation of EM emissions and
electrical currents observed either during seismic activity or in the laboratory
experiments. These include the electrokinetic and magnetohydrodynamic,
piezomagnetism, stress-induced variations in crustal conductivity, microfracturing,
and so on (Draganov et al., 1991; Park, 1996; Fenoglio et al., 1995; Egbert, 2002;
Simpson and Taflove, 2005). Whatever the physical mechanism of electromagnetic
generation is, it is well established that, during rock experiments conducted under
laboratory conditions, a strong electrical current is produced when rocks are stressed,
especially at the stage of the main rupture.

As the development of satellite Earth Observation (EO), there is an increasing
amount of evidence that during some last stages of the long term process of
preparation, there could be a transfer of energy between lithosphere, atmosphere and
ionosphere, so as to introduce the concept of a lithosphere–atmosphere–ionosphere
coupling (LAIC) among the three involved layers of the Earth (Pulinets et al., 1994,

2000; Hayakawa and Molchanov, 2002; Molchanov et al., 2004; Kamogawa, 2006).

When we investigate electromagnetic emissions induced by an electrical current or a magnetic moment on the surface or beneath the Earth, the effect of the medium air, crustal as well as ionosphere should be taken into account because of these three media being of different conductivities and so we need to consider a lithosphere-atmosphere-ionosphere electromagnetic coupling (Cummer, 2000).

Several tentative LAIC models have been constructed based on ground-based and ionospheric observations prior to strong earthquakes and the investigation of influence of external electrical filed on ionospheric parameters has been developed quickly (Pulinets and Ouzounov, 2011; Pulinets and Davidenko, 2014; Sorokin and Hayakawa,

2013; Sorokin and Hayakawa, 2014; Kuo et al., 2011, 2014; Namgaladze et al., 2012;

Zolotov et al., 2012; Zolotov, 2015). At the same time, the ionosphere plays an important role in electromagnetic propagation at Extremely Low Frequency (ELF)

and Very Low Frequency (VLF), the ground and the ionosphere are good electrical conductors and form a spherical Earth-ionosphere waveguide (Cummer, 2000). In addition, in the Controlled Source Electromagnetic (CSEM) method, widely used in petroleum exploration or mining, the ionospheric influence on electromagnetic (EM)

fields should be considered when the distance between a large-scale and large-power fixed source and the receiver is up to one thousand kilometers. EM fields can be amplified in the ionosphere as it is shown when we use analytical solutions of

Maxwell equations, as well as numerical ones of the "Earth-ionosphere" mode with a source on the Earth's surface or in the lower atmosphere (Fu et al., 2012; Li et al.,

2010a; Li et al., 2010b; Xu et al., 2012; Li et al., 2011).

Therefore, comparing with an electromagnetic attenuation without ionospheric effect, the point is to evaluate the ionospheric influence on the electromagnetic propagation when the distance between the epicenter and the observing location is up to one thousand kilometers or even more. Furthermore, the comparison between the observation distance reported by Li et al. (2013) (D=1,440 km) and the length of the

Wenchuan earthquake main rupture L=∼150 km (Zhang et al., 2009) indicates that the length of the dipole source is not negligible. So in this paper, based on the work of

Key (2009), a three-layer (Earth-air-ionosphere) physical model, as well as a two-layer (Earth-air) model, containing a finite length dipole current source co-located along the fault and beneath the Earth is introduced in Sect. 2. For specified parameters, some simulation results of the current source with and without ionospheric effect are given in Sect. 3. In Sect.4, using assumed parameters, the
simulation results for the case of the Wenchuan earthquake reported by Li et al. (2013)

[revised manuscript text omitted]
 were improved effectively and it would be easier to record a signal even at stations located beyond their detectability threshold.

[Figure]

**Fig.5.** The Wenchuan source producing electric field |Ex| decay curves as a function of the distance along x-axial direction with ionospheric effect (blue dot line), as well as without ionospheric effect (red line). The electric field |Ex|=1.5 mV m$^{-1}$ is labeled by a red arrow and a blue one respectively.

**4.3. Wave 2-D distribution**

We perform electromagnetic wave fields for the Wenchuan source and this is done in the ground plane region -1,000 km<x<1,000 km and -1,000 km<y<1,000 km in order to visualize the 2-D distribution of the wave power surrounding the electrical source.

Figure 6 displays the 2-D power distributions of the electrical field components |Ex|, |Ey| and the total |E| ($|E|^2=|Ex|^2+|Ey|^2$) after making a logarithm calculation on the Earth's surface. It can be seen firstly from Figure 6a that there is an obvious constant strong power along the current element length (-75 km<x<75 km) in the x-direction. The electrical value in this area is not discussed here because it is usually considered not precise. Then the strong field radiates outward surrounding four main axes, indicating 1 order rough decay of the field at ～160 km, 2 orders of magnitude at ～320 km from the source endpoint in the x-direction. There is only 3 orders decay till 1,000 km away because of the ionospheric facilitating effect on the field and it keeps a strong value (～1.86 mV) which can be fairly recorded by the stations. However, there are also weak power areas along lines, which form 45 °angle with the principal axis for the electrical field power |Ex| (Figure 6a). Complementally, the electrical field power |Ey| (Figure 6b) is basically characterized by strong power areas between two main axes, as well as weak ones along four chief axes. The power distribution of the total |E| consequently presents to be symmetry to the center circle outside of the source (Figure 6c), which also indicates that the radiating patterns of the electrical field power |Ex| and the electrical field power |Ey| are complementary (One is strong area and the other is weak area) each other surrounding the source.

[Figure]

(a)            (b)

(c)

**Fig.6.** 2-D distributions of electrical field power |Ex| (a), |Ey| (b) and total |E| (c) after a logarithm calculation for the Wenchuan source using Earth-air-ionosphere model.

**5 Discussion**

In very recent years, there is an increasing amount of evidence that during some last stages of the long term process of preparation, there could be a transfer of energy between lithosphere and the above layers of atmosphere and ionosphere, so as to introduce the concept of a lithosphere–atmosphere–ionosphere coupling (LAIC) among the three involved layers of the Earth system (Pulinets et al., 2000; Hayakawa and Molchanov, 2002; Molchanov et al., 2004; Pulinets and Ouzounov, 2011). On one hand, the 'energy source' is usually thought to be beneath the Earth's surface and related to tectonic activities in the lithosphere. On the other hand, numerous rock-pressure experiments and electromagnetic observations associated with seismic activities have already proved that a giant electrical current and an abrupt increase of electromagnetic signals occur during the main rupture of stressed-rocks. These phenomena happed on May 9 2008, 3 days before the Wenchuan event, which hypocenter lies in mid-crust. The strong seismo-telluric current is thought to run mainly along the Longmenshan fault and electromagnetic oscillations, induced by the current and predominated by ULF frequency band, propagate up to ionosphere and give rise to perturbations of ionospheric parameters. Some of these parameters have been investigated, such as GPS TEC and f0F2 (Yu et al.,2009; Xu et al., 2010; Akhoondzadeh et al., 2010), DEMETER satellite O+ density (Zhang et al., 2009), electron density and electron temperature (Zeng et al., 2009), and so on. Fortunately, all these study results present a climax on May 9 and this indicates a lithosphere–atmosphere–ionosphere coupling or interaction aroused by these electromagnetic signals prior to the Wenchuan event.

Unfortunately, at present, most of investigations put emphases on the effect of earthquakes upon the ionosphere and few of them pay attention to an inverse problem, that is the ionospheric influence on the electromagnetic waves passing through.

The ionosphere, as a part of the electrical conducting region of the upper atmosphere, can enhance electromagnetic fields and cause the decay as a function of distance to slow down when an observation is within ionospheric range and the ionosperic effect can be up to 1-2 magnitudes of the electrical fields in our simply three-layer model for some specified parameters we have selected here.

Considering the Wenchuan event, the electrical signals from the lithosphere interact with the ionosphere and are at the same time enhanced, and then registered at 1440 km Gaobeidian station with the amplitude of 1.3 mV m$^{-1}$. This electrical field is used to simulate the seismo-telluric current produced by the Wenchuan main rupture in an Earth-air-ionosphere model together with an Earth-air model. The results present that, the seismo-telluric currents with and without ionospheric effect must be about $5.0 \times 10^7$ A and $3.7 \times 10^8$ A respectively. Compared with the expected seismo-telluric current ~10–100 kA of the "Alum Rock" $M_W$=5.6 earthquake for an observed 30 nT pulse at 1 Hz and D=2 km (Bortnik et al., 2010), this result is probably in a reasonable range.

However, firstly, the total rupture of the Longmenshan fault during the
Wenchuan main shock is extremely complicated that comprises of tenths of rupture
stages and several pauses, totaling 90 s for the whole rupture process ($\sim$300 km),
according to Zhang et al.,(2009). Thus the total surface rupture $\sim$ 300 km is
nevertheless not used here. While performing the analysis on only the primary 30 s, a
main stage of the Wenchuan earthquake, out of 90 s as we have selected L=150 km
above, is expected to be representative of the majority of the rupture to generate a
seismo-telluric current. Secondly, three medium are thought of as a homogeneous
isotropic medium in our models and with the same average conductivity value for
each one, especially for the wenchuan area. However, the Earth conductivity plays
such an important role that it predominately affects the fluctuations of the electrical
fields as shown in Fig.4 although no one exactly knows the right conductivity of the
Earth medium at the rupture depth. The value $\sigma_1 = 7.0 \times 10^{-4}\, \mathrm{S\, m^{-1}}$ taken part in
all analysis is estimated when the observing frequency range f=0.1-10 Hz and the
hypocenter depth d=19 km of the Wenchuan main event are taken into account for the
skin-depth formula. One must also mention that we use f=1 Hz in our calculations
because we cannot identify the actual frequencies in the recorded analog signals. All
these can probably underscore our simulation results.

While these disadvantageous selections maybe are not so important at the same
time because the key point of this paper is of the ionospheric influence on
electromagnetic wave propagation and our investigation attains advantageous results.

The "selectivity" or "orientation" of the electromagnetic information is a very
important character during seismic activities (Varotsos and Lazaridou, 1991). For a
finite length dipole source of the Wenchuan earthquake, its 2-D distributions of
electrical field component |Ex| and |Ey| , which are orthogonal each other, on the
Earth's surface shows there are strong field power areas and weak field power areas
around the source as illustrated by [Bortnik et al., 2010]. While the radiating pattern
of the total |E| in this investigation is symmetry to the center circle outside of the
source which indicates a signal is always registered to anyone direction if a system is
designed to measure the total field |E| or both of |Ex| and |Ey| components instead of
only one. This result also basically supports the practices of "selectivity" or
"orientation", the observing reality before the Wenchuan earthquake described by Li
et al.[2013], for example, 'Compared with the EW (East-West) orientation, the electromagnetic signal is more obvious in the SN (South-North) orientation'. The
selectivity effect is a complex phenomenon that may be attributed to a superposition
of the following three factors: "source characteristics", "travel path" and
"inhomogeneities close to the station" [Varotsos and Lazaridou, 1991; Varotsos et al.,
2005]. Analytical solutions of Maxwell equations [Varotsos et al., 2000], as well as
numerical ones [Sarlis et al., 1999], convince that selectivity results from the fact that
earthquakes occur by slip on faults which are appreciably more conductive than the
surrounding medium.

**6 Conclusions**

[revised manuscript text omitted]

Freund, F., and Wengeler, H.: The infrared spectrum of OH⁻ compensated defect sites in C-doped MgO and CaO single crystals. J. Phys. Chem. Solids 43, 129–145,

1982.

Freund, F.: Charge generation and propagation in igneous rocks, J. Geodynamics, 33,

543–570, 2002.

Freund, F.: Conversion of dissolved "water" into molecular hydrogen and peroxy linkages. J. Non-Cryst. Solids 71, 195–202, 1985.

Freund, F.: Stress-activated positive hole charge carriers in rocks and the generation of pre-earthquake signals, in: Electromagnetic Phenomena Associated with

Earthquakes, edited by: Hayakawa, M., Transworld Research Network, Trivandrum,

India, Chapter 3, 41–96, 2009.

Freund, F.: Toward a unified solid state theory for pre-earthquake signals, Acta

Geophys., 58(5), 719–766, 2010.

Fu, C. M., Di, Q. Y., Xu, C., and Wang, M. Y.: Electromagnetic fields for different type sources with effect of the ionosphere, Chinese J. Geophys., 55(12), 3958–3968, doi: 10. 6038/ j. issn. 0001-5733. 2012. 12. 008, 2012(in Chinese with English abstract).

Guan, H. P., Han, F.Y., Xiao, W. J., and Chen, Z.Y.: ULF electromagnetic observation and data processing methods, Earthquake, 23(2), 5–93, 2003(in

Chinese with English abstract).

Hayakawa, M., and Molchanov, O. A. (Eds.): Seismo-Electromagnetics:

Lithosphere-Atmosphere-Ionosphere Coupling, Tokyo, Japan: TERRAPUB, 2002.

Kamogawa, M.: Pre-seismic lithosphere–atmosphere–ionosphere coupling. Eos

87(40), 2006.

Key, K.: 1D inversion of multicomponent, multi-frequency marine CSEM data:

Methodology and synthetic studies for resolving thin resistive layers, Geophysics,

74(2), F9–F20, 2009.

Kopytenko, Y. A., Matiashvili, T. G., Voronov, P. M., Kopytenko, E. A., and

Molchanov, O. A.: Detection of ultra-low frequency emissions connected with the

Spitak earthquake and its aftershock activity, based on geomagnetic pulsations data at Dusheti and Vardzia observatories, Phys. Earth Planet. Interiors, 77, 85–95, 1993.

Kuo, C. L., Huba, J. D., Joyce, G., and Lee, L. C.: Ionosphere plasma bubbles and density variations induced by pre-earthquake rock currents and associated surface charges. J. Geophys. Res., 116, A10317, 2011.

Kuo, C. L., Lee, L. C., and Huba, J. D.: An improved coupling model for the lithosphere-atmosphere-ionosphere system, J. Geophys. Res. Space Physics, 119(4), 3189–3205, 2014.

Li, M., Lu, J., Parrot, M., Tan, H., and Zhang, X.: Review of unprecedented ULF electromagnetic anomalous emissions possibly related to the Wenchuan $M_S = 8.0$ earthquake,on 12 May 2008.Nat.Hazards Earth Syst.Sci., 13(2), 279–286, doi: 10. 5194/nhess-13-279-2013, 2013.

Li, D., Di, Q. Y., and Wang, M. Y.: One-dimensional electromagnetic fields forward modeling for "earth–ionosphere" mode. Chinese J. Geophys., 54(9), 2375–2388, doi: 10. 3969/ j. issn. 0001–5733. 2011. 09. 021, 2011 (in Chinese with English abstract).

Li, Y., Lin, P. R., Zheng, C. J., Shi, F. S., Xu, B. L., and Guo, P.: The electromagnetic response modeling of the ELF method and the influence of the ionosphere, Geophysical & Geochemical Exploration, 34(3), 332–339, 2010a, (in Chinese with English abstract).

Li, D. Q., Di, Q. Y., and Wang, M. Y.: Study of large scale large power control source electromagnetic with "Earth–ionosphere" mode, Chinese J. Geophys., 53(2), 411–420, doi: 10. 3969/ j. issn. 0001-5733. 2010. 02. 019, 2010b, (in Chinese with English abstract).

Molchanov, O. A., Kopytenko, Y. A., Voronov, P. M., Kopytenko, E. A., Matiashvili, T. G., Fraser-Smith, A. C., and Bernardi, A.: Results of ULF Magnetic field measurements near the epicenters of the Spitak ($M$s 6.9) and Loma Prieta ($M$s7.1) earthquakes: comparative analysis, Geophys. Res. Lett., 19, 1495–1498, 1992.

Molchanov, O. A., Fedorov, E., Schekotov, A., Gordeev, E. , Chebrov , V., Surkov, V., …, Biagi, P. F.: Lithosphere-atmosphere-ionosphere coupling as governing mechanism for preseismic short-term events in atmosphere and ionosphere, Natural Hazards Earth Syst. Sci., 4, 757-767, 2004.

Namgaladze, A. A., Zolotov, O. V., Karpov, M. I., and Romanovskaya,Y.V.: Manifestations of the earthquake preparations in the ionosphere total electron content variations. Natural Science, 4(11), 848–855, 2012.

Ohta, K., Umeda, K., Watanabe, M. and Hayakawa, M.: Relationship between ELF magnetic field and Taiwan earthquake. In Lithosphere-Atmosphere-Ionosphere Coupling (eds M. Hayakawa and O. A. Molchanov), Terra Science Publishers, Tokyo, pp. 233–237, 2002.

Panfilov, A. A.: The results of experimental studies of VLF–ULF electromagnetic emission by rock samples due to mechanical action, Nat. Hazards Earth Syst. Sci., 14, 1383–1389, doi:10.5194/nhess-14-1383-2014, 2014.

Park, S. K.: Precursors to earthquakes: seismo-electromagnetic signals, Surv. Geophys., 17, 493–516,1996.

Pulinets, S. A., and Davidenko, D.: Ionospheric precursors of earthquakes and Global Electric Circuit. Advances in Space Research, 53(5), 709–723, 2014.

Pulinets, S. A., and Ouzounov, D.: Lithosphere-Atmosphere-Ionosphere Coupling (LAIC) model-An unified concept for earthquake precursors validation, J. Southeast Asian Earth Sci., 41(4–5): 371–382, 2011.

Pulinets, S. A., Boyarchuk, K. A., Hegai, V. V., Kim, V. P., and Lomonosov, A. M.: Quasielectrostatical model of atmosphere-thermosphere-ionosphere coupling, Adv. Space Res., 26, 1209-1218, 2000.

Pulinets, S.A., Legen'ka, A.D., Alekseev, V.A., 1994. Pre-earthquakes effects and their possible mechanisms. In: Dusty and Dirty Plasmas, Noise and Chaos in Space and in the Laboratory. Plenum Publishing, New York, pp. 545–557.

Qian, S., Hao, J., Zhou, J. and Gao, J.: Precursory Electric and Magnetic Signals at ULF and LF Bands during the Fracture of Rocks under Pressure. Earthquake Research in China, 19(2), 109–116, 2003 (in Chinese with English abstract).

Qian, S., Hao, J., Zhou, J. and Gao, J.: Simulating experimental study on ULF electromagnetic precursors before Jiji $M$s = 7.4 earthquake. In Lithosphere-Atmosphere-Ionosphere Coupling (eds Hayakawa, M. and Molchanov, O. A.), Terra Science Publishers, Tokyo, pp. 49–53, 2002.

Qian, S., Ren K., Lü, Z.: Experimental study on VLF, MF, HF and VHF electromagnetic radiation characteristics with the rock breaking, Earthquake Science, 18(3), 346–351, 1996 (in Chinese with English abstract).

Sarlis, N., Lazaridou, M., Kapiris, P., and Varotsos, P.: Numerical model of the selectivity effect and the V/L criterion, Geophys. Res. Lett., 26, 3245–3248, 1999.

Scoville, J., J. Sornette, and Freund, F. T.: "Paradox of peroxy defects and positive holes in rocks Part II: Outflow of electric currents from stressed rocks." Journal of Asian Earth Sciences 114, Part 2: 338-351, 2015.

Simpson, J. J., and Taflove, A.: Electrokinetic effect of the Loma Prieta earthquake calculated by an entire-Earth FDTD solution of Maxwell's equations. Geophys. Res. Lett., 32, L09302, doi: 10. 1029/2005GL022601, 2005.

Sorokin, V. M., and Hayakawa, M.: Generation of Seismic-Related DC Electric Fields and Lithosphere-Atmosphere-Ionosphere Coupling. Modern Applied Science, 7(6), 1–25, 2013.

Sorokin, V. M., and Hayakawa, M.: Plasma and Electromagnetic Effects Caused by the Seismic-Related Disturbances of Electric Current in the Global Circuit. Modern Applied Science, 8(4), 61–83, 2014.

Uyeda, S., Nagao, T., Orihara, Y., Yamaguchi, T., and Takahashi I.: Geoelectric potential changes: Possible precursors to earthquakes in Japan, Proc. Nat. Acad. Sci., 97, 4561–4566, 2000.

Varotsos, P., and Lazaridou, M.: Latest aspects of earthquake prediction in Greece based on seismic electric signals. Tectonophysics, 188, 321–347,1991.

Varotsos, P., Sarlis, N., and Lazaridou, M.: Transmission of stress induced electric signals in dielectric media, Part II, Acta Geophys, 48, 141–177, 2000.

Varotsos, P., Sarlis, N., Skordas, E., Tanaka, H., and Lazaridou, M.: Additional evidence on some relationship between seismic electric signals and earthquake source parameters, Acta Geophys., 53, 293–298, 2005.

Wait, J. R.: Geo-electromagnetism: Academic Press,1982.

Wait, J. R.: Some Factors Concerning Electromagnetic Wave Propagation in the Earth's Crust, Proc. IEEE, 54(8), August 1966.

Xu, C., Di, Q. Y., Fu, C. M. and Wang, M. Y.: The contrast of response characteristics between large power long dipole and circle source, Chinese J. Geophys, 55(6), 2097–2104, doi: 10. 6038/ j. issn. 0001–5733. 2012. 06. 03, 2012, (in Chinese with English abstract).

Xu, T., Hu, Y., Wu, J., Wu, Z., Suo, Y., and Feng, J.: Giant disturbance in the ionospheric F2 region prior to the $M$8.0 Wenchuan earthquake on 12 May 2008, Ann. Geophys., 28, 1533–1538, 2010.

Xu, X. W.:Album of 5.12 Wenchuan 8.0 earthquake surface ruptures. Seismological press, 2009 (in Chinese with English abstract).

Yamauchi, T., Maekawa, S., Horie, T., Hayakawa, M., and Soloviev, O.: Subionospheric VLF/LF monitoring of ionospheric perturbations for the 2004 Mid-Niigata earthquake and their structure and dynamics, J. Atmos. Sol. Terr. Phys., 69, 793–802, 2007.

Yu, T., Mao, T., Wang, Y. G., and Wang, J. S.: Study of the ionospheric anomaly before the Wenchuan earthquake, Chinese Science Bulletin, 54(6): 1086–1092, doi: 10.1007/s11434-008-0587-8, 2009 (in Chinese with English abstract).

Zeng, Z. C., Zhang, B., Fang, G. Y., Wang, D. F., and Yin, H. J.: The analysis of ionospheric variations before Wenchuan earthquake with DEMETER data, Chinese J. Geophys., 52(1): 11–19, 2009 (in Chinese with English abstract).

Zhang, X., Shen, X., Liu, J., Ouyang, X., Qian, J., and Zhao, S.: Analysis of ionospheric plasma perturbations before Wenchuan earthquake. Nat. Hazards Earth Syst. Sci., 9: 1259–1266, 2009.

Zhang ,Y., Feng, W. P., Xu, L. S., Zhou, C. H., and Chen, Y. T.: Spatio-temporal rupture process of the 2008 great Wenchuan earthquake, Science in China Series D: Earth Sciences, 52 (2), 145–154, 2009.

Zhu, Y. T., Wang, X. B., Yu, N., Gao, S. Q., Li, K., and Shi, Y. J.: Longmenshan
magnetotelluric deep structure and the Wenchuan earthquake ($M_S$8.0), Acta
Geologica Sinica, 82 (12), 1769–777, 2008 (in Chinese with English abstract).
Zolotov, O. V.: Ionosphere Quasistatic Electric Fields Disturbances over Seismically
Active Regions as Inferred from Satellite_Based Observations: A Review. Russian
Journal of Physical Chemistry B, 9(5), 85–788, 2015.
Zolotov, O. V., Namgaladze, A. A., Zakharenkova, I. E., Martynenko, O. V.,
andShagimuratov, I. I.: Physical Interpretation and Mathematical Simulation of
Ionospheric Precursors of Earthquakes at Midlatitudes. Geomagnetism &
Aeronomy, 52(3), 390–397, 2012.